# Nurses', patients', and informal caregivers' attitudes toward aggression in psychiatric hospitals: A comparative survey study

Maritta Välimäki[1¤a¤b]*, Joyce Lam[1¤a], Daniel Bressington[1¤c], Teris Cheung[1], Wai Kit Wong[1¤d], Po Yee Ivy Cheng[2], Chi Fai Ng[3], Tony Ng[4], Chun Pong Yam[5], Glendy Ip[6,7], Lee Paul[1¤e], Tella Lantta[¤a¤f]

1 School of Nursing, Hong Kong Polytechnic University, Hung Hom, Kowloon, Hong Kong (SAR), China, 2 Community Psychiatric Services, Pamela Youde Nethersole Eastern Hospital, Chai Wan, Hong Kong (SAR), China, 3 Department of Psychiatry, Tai Po Hospital, Tai Po, Hong Kong (SAR), China, 4 Department of Psychiatry, Pamela Youde Nethersole Eastern Hospital, Chai Wan, Hong Kong (SAR), China, 5 Department of Psychiatry, Kowloon Hospital, Kowloon, Hong Kong (SAR), China, 6 Central Nursing Division, Kwai Chung Hospital, Kwai Chung, Hong Kong (SAR), China, 7 Hong Kong College of Mental Health Nursing, The Hong Kong Academy of Nursing, Hong Kong (SAR), China

¤a Current address: Department of Nursing Science, University of Turku, Turku, Southwest Finland, Finland
¤b Current address: Xiangya School of Nursing and Xiangya Research Center of Evidence-Based Healthcare, Central South University, Changsha, Hunan, China
¤c Current address: College of Nursing and Midwifery, Charles Darwin University, Darwin, Northern Territory, Australia
¤d Current address: School of Nursing, Tung Wah College, Homantin, Kowloon, Hong Kong (SAR), China
¤e Current address: University of Leicester, Department of Health Sciences, College of Life Sciences, United Kingdom
¤f Current address: Department of Nursing, Faculty of Health and Education, Manchester Metropolitan University, Manchester, United Kingdom
* maritta.vaelimaeki@csu.edu.cn

**Data Availability Statement:** The basic information of the study and the study dataset are

## Abstract

Attitudes toward aggression is a controversial phenomenon in psychiatry. This study examined and compared attitudes toward patient aggression in psychiatric hospitals from the perspectives of nurses, patients and informal caregivers and identified factors associated to these attitudes. A total of 2,424 participants completed a self-reported instrument regarding attitudes toward aggression (12-items Perception of Aggression Scale; POAS-S). We analysed data from nurses (n = 782), patients (n = 886), and informal caregivers (n = 765). Pearson's *r* correlations were used to examine associations between variables. Differences between group scores were analysed using ANOVA/MANOVA with post-hoc Sheffe tests. Multivariate logistic regression models and logistic regression analysis were used to examine the effects of respondents' characteristics on their attitudes toward aggression. Nurses had significantly more negative and less tolerant perceptions toward aggression (mean [SD] 47.1 [7.5], p<0.001) than the patients (mean [SD] 44.4 [8.2]) and the informal caregivers (mean [SD] 45.0 [6.9], according to the POAS-S total scores. The same trend was found with the dysfunction and function sub-scores (mean [SD] 25.3 [4.1] and 15.0 [3.6], respectively); the differences between the groups were statistically significant (p <0.001) when nurses' scores were compared to those of both the patients (mean [SD] 23.7 [5.3] and 14.0 [4.1], respectively) and the informal caregivers (mean [SD] 24.4 [4.2] and 13.9 [3.5],

available at doi.org/10.17504/protocols.io.
j8nlkk62wl5r/v1.

**Funding:** Prof Dr. Maritta Välimäki has received
funding for this study (Start-up funding by the
Hong Kong Polytechnic University, 1-ZE84, https://
www.polyu.edu.hk/sn/research/research-projects/
and the Academy of Finland fund (grant numbers:
294298, 307367, https://www.aka.fi/en/]). The
funders had no role in study design, data collection
and analysis, decision to publish, or preparation of
the manuscript.

**Competing interests:** The authors have declared
that no competing interests exist.

respectively). The study offers new understanding of aggressive behavior in different treat-
ment settings where attitudes toward patient behavior raises ethical and practical dilemmas.
These results indicate a need for more targeted on-the-job training for nursing staff, aggres-
sion management rehabilitation programs for patients, and peer-support programs for infor-
mal caregivers focused on patient aggression.

## Background

The association between increased risk of violent behavior and serious mental disorders has
been documented, although contradictory opinions exist [1]. Schizophrenia and other psy-
chotic conditions are associated with a high frequency of violent behavior [2–5]. Most violent
events occur in hospital institutions [6, 7]. A systematic review and meta-analysis by Iozzino
et al. [4]. showed that almost 20% of patients admitted to acute psychiatric care behave vio-
lently. In the United States, 25–85% of health care workers have experienced physical violence
during the past year [8], while 8–38% of health care workers globally are assaulted at least once
during their career [9].

Research on factors related to patient aggression in psychiatric hospital settings typically
focuses on patient characteristics. Being younger, male, involuntarily admitted, single, having
a diagnosis of schizophrenia and a greater number of previous admissions, and having a his-
tory of violence, self-destructive behavior or substance abuse all increase the risk of violent
behavior [2]. Discussion related to patient aggression has also been extended to factors beyond
patient characteristics [10]. For example, personal characteristics, the quality of interaction
between patients and staff, as well as organizational and environmental variables such as staff/
patient ratio, how care has been organized, ward policy, staff morale, nurses' educational level,
clinical experience [11] or staff gender [12, 13] may also contribute to patient aggressive
behavior.

On the other hand, aggression is a complex process, which integrates social and cognitive
aspects of those persons who are involved, affected by a combination of their personality, indi-
vidual traits, situation, and decision-making processes [14]. Attitudes toward aggression plays
a crucial role in how aggressive incidents occur, are seen, and are managed in healthcare set-
tings [15]. Weltens et al. [16] described that development and expression of aggression is a
multifactorial event, which includes attitudes toward patients, cultural factors, and clinician-
related factors such as attitudes toward aggression. McCann et al. [17] concluded, based on
their study, that attitudes toward the management of aggression are complex and contradic-
tory: staff's attitudes can affect the way they manage aggression, and therefore, a wide range of
initiatives are needed to prevent and deal with patient challenging behavior. Recent studies
have also described the association between attitudes and practices. Efkemann et al. [18], for
example, confirmed, using a vignette study, that a general approving attitude toward coercion
significantly influenced decisions around coercion in individual cases and resulted in a more
likely approval of applying coercion.

Abderhalden et al. [19] found two dimensions of nurses' perceptions of aggression: they see
it as a dysfunctional/undesirable event, but also as a functional/comprehensible phenomenon
not solely viewed as negative. Verhaeghe et al. [13] further summarized three different per-
spectives on attitudes toward patient aggression. First, aggression can be seen as a harmful and
dysfunctional phenomenon. Second, aggression as a functional phenomenon is a way to
express ones needs. The third dimension characterizes aggression as a normal phenomenon—

a reaction to feeling anger. Depending on how one sees the phenomenon, reactions toward patient aggression may vary. Bowers [20] further emphasized that attitudes have an important role in aggressive events in psychiatric services: nurses with positive attitudes toward personal disorders are better able to manage their own emotional reactions, particularly to patient violent behavior, and are more likely to defuse tense situations and turn the conflict into a therapeutic opportunity. Likewise, restrictive attitudes among staff may provoke aggressive incidents and lead to an increase in the use of coercive measures [21].

On the other hand, attitudes toward patient aggression are not stable and may vary depending on the context, such as country [22] or the type of ward where the aggression occurs [15]. Australian nurses in old age psychiatric units were found to hold positive attitudes toward aggression although they were pessimistic toward the management of aggression, thinking that aggression is inevitable in psychiatry settings [17]. In Belgium [13] and Sweden [23], males and more experienced nurses were more likely to blame patients for their aggressive behavior. On the other hand, Whittington [24] reported in England that experienced staff members had more tolerant attitudes toward patient aggression. Female nursing students in Spain [25] and nursing students in Turkey [26] commonly considered aggression to be unacceptable. Nursing students in England who viewed aggression as unacceptable were found to be less likely to have a positive overall attitude to personality disorder and were unlikely to accept these patients [27].

Nurses' attitudes toward aggression have been widely studied worldwide, and the research has added valuable knowledge in mental health care. However, research is still scarce on patients' and informal caregivers' attitudes toward aggression, although they represent an important network around persons with mental health problems. Studies have suggested that families in the home environment generally have little knowledge about aggressive behavior and their coping skills to manage aggression are limited [28, 29]. A survey among Chinese caregivers of people with severe mental illness revealed that attitudes toward patient violent behavior are rather pessimistic [29].

The limited focus on attitudes toward aggression may overlook intersubjective opinions that could warrant dialogue between nurses, patients, and informal caregivers. As patients themselves and their informal caregivers are most directly affected by the outcomes of patients' aggressive behavior, knowledge of patients' and their informal carers' attitudes of aggression should be of pivotal interest. To identify a possible knowledge gap, we systematically searched comparative cohort studies concerning attitudes toward aggression among nurses, patients, and informal caregivers. We found four studies, all of which were limited to comparisons on attitudes toward aggression between nurses and patients only. The results were contradictory. In a study by Gillig et al. [30], the majority of both nursing staff and patients in a psychiatric unit agreed that patients who were psychotic were more likely to be involved in incidents of physical aggression. More staff, but fewer patients, attributed aggressive incidents to altered cognitive function and diminished impulse control. At the same time, a large number of staff believed that physical aggressive episodes are learned behaviors or are related to a patient's feelings or beliefs. In a study by Duxbury and Whittington [31], patients perceived that aggressive behavior is due to external issues, restrictive environments on the wards, and poor communication with staff, while staff saw the patient's illness as the main cause of aggressive behavior. In another study, Dickens et al. [32] reported that, compared to staff, patients had more positive attitudes toward the possibility of violence prevention due to the modifiable nature of aggression. On the contrary, a survey conducted in a highly secure setting showed that the staff's and patients' views about patient aggression were similar overall [33].

Due to contradictory study results, no clear conclusion about the differences or similarities in attitudes toward aggression among various stakeholder groups can be derived. Knowledge

about attitudes toward patients is important as, globally, this is one of the key issues in psychiatric hospitals. The majority of studies focus primarily on the contribution of patient factors in developing aggression, neglecting ward and staff factors that may be more promising targets for interventions aiming to reduce or prevent aggression development on inpatient psychiatric wards [19]. Identification of possible similarities or differences in attitudes can contribute to an increased understanding of different parties who have witnessed patient aggressive behavior in hospital settings. Possible differences in the understanding of treatment situations may result in ineffective medical care, patient and family dissatisfaction, and distress [34]. Although differences in attitudes toward aggression may not be a problem per-se, it is important to be aware of these variations and to better understand the factors that account for different opinions [16]. Therefore, comparisons of different perspectives about attitudes toward aggression are crucial for effective aggression management [31]. Based on this knowledge, targeted interventions and training could be used to promote humane ways to better manage patient aggression and relieve burden caused by aggressive events [29, 32, 35].

It Is also of paramount importance to better understand attitudes toward patient aggression in Asia. Many deficits in Asian countries still exist in the care of people with mental illnesses [36] and stigmatized attitudes among the public and professionals seem to change slowly in these settings [36, 37]. Being aware of attitudes toward patient aggression from different angels can open new insight into patient care, not only in psychiatric services but also in a variety of clinical settings where staff members handle patient aggressive behavior on a daily basis.

## Materials and methods

### Aim

To describe and compare attitudes toward patient aggression in psychiatric hospitals among nurses, patients and informal caregivers and to identify factors associated with these attitudes.

### Design

A cross-sectional survey was conducted using a self-reported structured questionnaire in Hong Kong (China, SAR). The survey method was most appropriate for our purposes as the same structured items for all participants made it possible to compare perceptions about the phenomenon. In addition, as aggression is a sensitive topic, a survey allowed participants to express their views anonymously.

### Setting

The data were collected on Hong Kong inpatient hospital wards, which offer psychiatric services to patients in specific geographical clusters. Psychiatric services in Hong Kong include inpatient, outpatient, and community services. Inpatient services are divided into clinical specialties, for example, the Child and Adolescent Psychiatric Service, the Adult Psychiatric Service, the Psychogeriatric Service, the Substance Abuse Assessment Unit and the Psychiatric Unit for Learning Disabilities [38]. In 2019, seven hospital clusters with 3,647 psychiatric beds served the Hong Kong population. Between 1 Apr 2018 and 31 March 2019, a total of 18,501 discharges were reported [39].

All seven hospital clusters were invited to join the study during a cluster meeting of Nurse Directors (7 February 2017). Five hospitals in five separate clusters were willing to join the study. Before application of the ethical assessment of the study, one hospital withdrew, which left us with four hospitals. At the time of the data collection, about 1,400 nurses (qualified/non-qualified) were working in the four study hospitals. (Table 1).

**Table 1. Hospital characteristics (April 2017).**

|  | Hospital A | Hospital B | Hospital C | Hospital D |
|---|---|---|---|---|
| Total number of nurses in the hospital | 521 | 230 | 182 | 95 |
| Number of wards using physical seclusion | 8 | 4 | 4 | 6 |
| Total number of patient beds | 400 | 192 | 180 | 250 |
| Occupancy rate of the beds | 70–90% | 70–90% | 70–90% | 50–70% |
| Total number of patient admissions | 3,988 | 1,822 | 2,927 | 2,438 |

Nurse Directors in the four hospitals invited all the adult inpatient wards where coercive methods were used to manage patient aggression to join the study. Psychiatric outpatient departments, psychogeriatric inpatient units, and units with no direct inpatient care were not included as study wards.

## Sampling method

Convenience sampling was used for the data collection for all participant groups. The method was usable for our study as, due to privacy reasons, we were not able to record any identifiable information of informal caregivers visiting the study wards during the data collection period.

## Eligibility criteria

**Nurses.**   The inclusion criteria for nurses were as follows: registered nurses in psychiatry (Registered Nurse, Psychiatric, RNP), enrolled nurses (Enrolled Nurse, Psychiatric, ENP), and supporting staff without registration or enrolment in the Nursing Council of HK (Health Care Assistant, HCA; and Patient Care Assistant, PCA). All of these groups are potential victims of aggression in the same settings and have regular aggression training at the hospital (Level I: one hour online lecture about the causes, signs, symptoms, and the model of violence, and situation awareness; Level II: half-day workshop about how to disengage when a nurse is attacked; Level III: 1.5 day workshop about control and restraint techniques including team formation, holding skills etc.). Requested by the Occupational Safety Department by the Hospital Authority, Level I training is compulsory for all frontline staff, while Levels II and III are compulsory for those nurses working in psychiatric settings. The Committee of the Hospital Authority has also recommended that Levels I and II should be updated every year. Nursing staff working in psychiatric settings need to complete all three levels in one year. In addition, as the inclusion criteria for nurses, it was assumed that staff were working full time (44 hours/week) at the time of the data collection, and that participation was voluntary.

Allied health care professionals (e.g. physicians, occupational therapists, physiotherapists, social workers), nursing staff without direct patient care, those who were not available during the data collection period due to long-term sick leave, study leave, and those working on a part-time basis only were excluded.

**Patients.**   Patients invited to join the study were 18 years old or above, able to read and speak Cantonese, mentally stable (evaluated by the case medical officer), willing to join the study based on their free will (signed informed consent form), and in the discharge process (the discharge day could be identified in hospital medical records) to ensure that each patient's mental status was stable enough for them to participate in the survey. We excluded patients under physical restraint, those who were secluded or sedated, and those who did not sign the written consent form.

**Informal caregivers.**   Informal caregivers, defined as unpaid individuals (a spouse, partner, family member, friend or neighbor) involved in assisting others with activities of daily

living and/or medical tasks, were invited to participate in the study [40]. The recruitment occurred during their visit on the study ward at the time of the data collection period (the patient-family member ratio 1:1). The inclusion criteria were age 18 years or older, able to read and speak Cantonese, and willing to participate in the survey.

## Sample size calculation

Based on power calculations, 384 participants per group were needed to describe and compare nurses' and patients' attitudes toward aggression in psychiatric hospitals. With this group size, we could achieve a 5% margin error with a level of significance of 0.05. For nurses, we expected a response rate of 47% [35]. We expected the response rate for patients to vary from 61% [41] to 82% [42], and for family members from 45% to 51% [43, 44]. Therefore, we assumed that we needed to approach about 730 nurses (response rate 53%), 532 patients (response rate 72%) and 802 family members (response rate 48%) to obtain the sample size needed for the statistical analysis.

## Instruments

Attitudes toward aggressive behavior were collected with the 12-item Perception of Aggression Scale (POAS-S) [45], which is a shortened version of the original scale (POAS) [46]. The POAS has been used in varying combinations of items, for example, in Switzerland [19, 47, 48], Sweden [23], the USA [49], France [50], Spain [25], Turkey [26, 51], Taiwan [52] and Poland [53]. In this study, we used the self-administered Cantonese version of POAS-S [35]. Each of the 12 items includes a different definition of aggression that can be variously endorsed or rejected by respondents using a 5-point Likert scale (1 = strongly disagree, 2 = disagree, 3 = no idea, 4 = agree, 5 = strongly agree). The total score was calculated based on the 12 items (6 revised items) with a range between 12 to 60: a higher score indicated more negative view or lower tolerance toward patient aggression. Three sub-scales were used based on previous studies [23, 50]: 1) aggression as a dysfunctional/undesirable phenomenon, 2) aggression as a positive expression, and 3) aggression as a protective measure. The Chinese version of the scale has satisfactory internal consistency (Cronbach's alpha = 0.76–0.83) and good test-retest reliability (Pearson's $r$ = 0.87) [35]. In this study, the Cronbach alpha value for the POAS-S scale was 0.69 for nurses, 0.78 for patients, and 0.62 for informal caregivers.

Background information of all participants were age, gender, education level, and whether they had been victim of a patient physical aggressive incident in the past 12 months (yes, no). Nurses identified their position in the hospital (Advance Practice Nurse, APN/Ward Manager, Registered Nurses, or supporting staff without nursing degree), the length of their clinical working experience, and any additional training for violence management in the past 12 months.

Information on patients' employment status (unemployment, comprehensive social security assistance, student, disability allowance, employed, or other) was recorded. Psychiatric diagnoses were collected as recorded in the medical records (DSM-IV-TR/DSM-V) using the following diagnosis categories [54]: neurodevelopmental disorder [intellectual disability, mental retardation, autism etc.], schizophrenia spectrum and other psychotic disorders [schizophrenia, schizoaffective, delusional disorder, other psychoses], affective/mood disorder [depressive disorders, bipolar disorders, anxiety disorders], and substance-related and addictive disorders and others [PTSD, eating disorders, somatoform, OCD, conduct disorder, personality disorders etc.], the number of previous hospitalizations, and aggression history [verbal abuse, physical violence, aggression toward objects, self-harm].

Informal caregivers were asked to give additional information about their employment status, any self-reported diagnoses and the history of aggressive behavior with the patient they visited.

## Recruitment and data collection

The data collection took place between December 2017 to July 2018. An ethical review of the study proposal was conducted by the Ethical Institutional Board from the Hong Kong Polytechnic University (the Human Subjects Research Ethics Committee; no HSEARS20170206007). The Human Ethics Board of each of the four hospital clusters of the Hospital Authority approved the study (HKECREC-2017-038; KC/KE-17-0113/FR-3; KW/FR-18-044(121–04); NTEC-207-0125).

For nurses, a mass email invitation was sent within each hospital, and posters were displayed on the walls of the study settings to increase awareness of the study. Ward meetings were also organized for staff members to share oral and written information about the study. A contact person was selected for each ward. A specific number of questionnaires with empty envelopes were distributed by contact persons to the wards for nurses to complete. Completed and returned questionnaires were used as evidence of nurses' implied consent to participate in the study; no written informed consent for nurses was requested.

On each study ward, patients' eligibility was screened by a contact person together with the psychiatrist. Patients who fulfilled the criteria and showed interest in participating received oral and written information about the study from Research Assistants (RAs) on a short leaflet. After oral and detailed written information was shared with patients, patients received an informed consent form and a questionnaire to be completed anonymously in a private place on the ward. If a patient had difficulty reading or filling in the form independently, an RA read the items to the participant, and the patient completed the form. Patients sealed the completed questionnaires in an envelope and put it into a closed box.

Informal caregivers were recruited to join the study by an RA during their visits on the wards. After a preliminary introduction of the study, its purpose and procedure were explained in more detail to the informal caregivers. The informal caregivers signed a consent form, filled in the survey forms in a quiet place, and returned them to the RA before they left the ward.

## Ethical issues

The global principles of research ethics were followed in each phase of the study [55]. The participants in all groups received written and oral information about the study (purpose, aims, goal of the survey). Practical arrangements of the study, ethical issues (possible benefits, voluntary participation, and the right to refuse to participate at any stage of the study without giving a reason) were described. The voluntariness and confidentiality of the study were explained. It was also emphasized that the participants had the right to refuse or withdraw from the study at any stage without consequences. After receiving information about the study, each participant had an opportunity to ask questions, and they were allowed to think about their participation in the study. The contact information of each study ward was shared if further questions were to arise. Surveys were anonymous to protect participants' privacy regarding this sensitive research topic. Each completed questionnaire was coded with identification numbers only. The participants' well-being and emotional reactions were followed in case the survey caused any distress or uncomfortable feelings. Three research members of the team (MV, JYL, PL) had full access to the data.

## Data analysis

To ensure high-quality data management, a randomly selected 10% sub-sample of the data (244 forms, 4,968 items) was checked; 9 mistakes were found and corrected (0.49%). The length of the nurses' working experience was categorized (0–5 years, 6–10 years, 11–15 years, 16–20 years, and over 20 years), and three groups of hospital positions were recategorizeds: nurses in a frontline position (RN/enrolled nurse), supporting staff, and nurses in a leading position.

Exploratory analysis (frequencies, percentages, Mean, SD, Mode, range) was conducted. Comparisons between groups for each item were calculated using Chi-square tests (nominal scale). Scores for the total scale and three sub-scales (aggression as a dysfunctional/undesirable phenomenon, 6 items; aggression as a positive expression, 4 items; and aggression as a protective measure, 2 items) were formed by summing the value of each item. Correlations between variables were examined using Pearson's $r$. The comparison between groups of total scores and subscale scores were analysed using ANOVA/MANOVA with a post-hoc Sheffe test using ordinal and interval scale parametrics. To examine if differences in attitudes toward aggression in the total sample and across the three different groups were associated with the respondents' characteristics, variables with significant group differences were entered into multivariate logistic regression models, and logistic regression analyses were performed, with attitudes toward aggression as dependent variables. Pairwise deletion was used to manage missing values (i.e. records without missing data were used in any particular analysis). To evaluate the goodness of fit of the logistic regression model and the power of explanation of the model, Nagelkerke's R squared was calculated. In addition, to reduce possible multicollinearity in the regression model, we identified the variables that were the most collinear, i.e. nurses' age and length of work experience. We then conducted a sensitivity analysis by removing the variable 'work experience' out of the regression model and then restudied the relationship between the dependent variable (attitude to aggression) using a single independent variable (age) in the regression model.

The data were analysed using SPSS version 25.0 for the Windows platform [56]. All tests were 2-tailed and p-values < .05 were considered statistically significant. Bonferoni corrections were used if multiple corrections were needed to minimize the risk of type I errors. Results for multivariate logistic regression analysis were presented in odds ratios (ORs) and 95% confidence intervals (CI).

# Results

## Characteristics of the participants

In total, 2,424 responses were analysed (nurses n = 782, patients n = 886, informal caregivers n = 765). Fewer than two-thirds of the nurse participants were in the youngest (< 30 years) or the oldest (51 years and older) age group. There was an equal number of females and males. Half of the participants had not graduated with at least a postgraduate degree. Twenty-eight percent of nurses had faced a violent incident during the last 12 months (Table 2). The biggest response group was of nurses (n = 650) in a frontline position (RN/enrolled nurse n = 382, 58.8%), followed by health care assistants (n = 181, 27.7%%), then nurses in a leading position (advanced practice nurses, ward managers, n = 87,13.5%). The length of work experience among nursing staff varied between 0–10 years (42.1%), 11–20 years (20.5%), and over 20 years (37.5%). Most had participated in aggression management training organized by the study organization in the past 12 months (Level I 88.1% [n = 659], Level II 83.6% [n = 625], Level III 74.0% [n = 554]).

**Table 2. Characteristics of the nurse, patient, and informal caregiver groups.**

| | Nurses | Patients | Informal caregivers |
|---|---|---|---|
| | N (%) | N (%) | N (%) |
| **Age, n = 2,403** | **N = 768** | **N = 883** | **N = 752** |
| < 30 | 223 (29.0) | 207 (23.4) | 102 (13.6) |
| 31–40 | 165 (21.5) | 220 (24.9) | 101 (13.4) |
| 41–50 | 163 (21.2) | 186 (21.1) | 134 (17.8) |
| 51 > | 217 (28.3) | 270 (30.6) | 415 (55.2) |
| **Gender, n = 2,391** | **N = 754** | **N = 884** | **N = 753** |
| Female | 382 (50.7) | 413 (46.7) | 445 (59.1) |
| Male | 372 (49.3) | 471 (53.3) | 308 (40.9) |
| **Education, n = 2,408** | **N = 772** | **N = 883** | **N = 753** |
| < Secondary | 83 (11.1) | 380 (43.1) | 306 (40.7) |
| High school | 172 (22.3) | 285 (32.3) | 229 (30.4) |
| Tertiary and vocational | 134 (17.4) | 115 (13.0) | 83 (11.0) |
| Undergraduate or more | 381 (49.4) | 103 (11.6) | 135 (17.9) |
| **Victim of aggression, n = 2,386** | **N = 750** | **N = 883** | **N = 753** |
| | 210 (28.0) | 187 (21.2) | 51 (6.8) |

In the patient group, the distribution of age was generally homogenous, but slightly higher in the age group over 51 years (30.7%). About half were male (53.3%). Typically, patients had secondary school education or less (43.0%). About one-fifth (21.2%) of patients had faced aggression from other patients during the last 12 months. (Table 2.) The most common diagnosis was schizophrenia (0.8% neurodevelopmental disorder, 45.3% schizophrenia spectrum and other psychotic disorder, 44.8% affective/mood disorder, 3.6% substance-related addictive disorder, 5.5% other, N = 727). The number of previous hospitalizations varied: one (30%), two (19%), three (11.8%), four or above (36%) (n = 865). One-third were unemployed (31.4%), or they were receiving comprehensive social security assistance (18.2%) or disability allowance (17.6%). The rest of the participants were students (5.5%), employed (19.1%), or other (16.8%) (n = 885).

Informal caregivers formed the oldest participant group (55.4% over 51 years), of which 59.1% were females (Table 2). A small majority (40.6%) had secondary school or less as their education level. Of the informal caregivers, 6.5% had faced a violent incidence during the last 12 months (Table 2). Informal caregivers visited patients whose diagnoses included neurodevelopmental disorder (4.8%), schizophrenia spectrum and other psychotic disorder (44.1%), affective/mood disorder (43.3%), substance-related addictive disorder (2.5%) or other (5.3%) (n = 682). Most were employed (42%) or belonged in the 'other' category, for example, they were receiving family financial support, they were a housewife etc. (36.9%); 9.8% were unemployed; 4.5% were receiving comprehensive social security assistance; 4.2% were students; and 2.0% were receiving disability allowance (n = 755).

## Comparison of attitudes toward aggressive behavior between nurses, patients, and informal caregivers

Out of 12 individual items, the exploratory analysis showed significant differences in eleven items concerning the percentages of attitudes toward aggression between three respondent groups. Nurses agreed more often than patients or informal caregivers that aggression is unpleasant and repulsive, is unnecessary and unacceptable, and hurts others mentally or

**Table 3. Differences between POAS-S total scores and sub-scores in nurses', patients', and informal caregivers' data.**

| Scores | N | Min | Mean | SD | Mode | Max | F-value | p |
|---|---|---|---|---|---|---|---|---|
| **Total score** | | | | | | | | |
| Nurses | 782 | 12 | 47.1 | 7.5 | 48 | 60 | 27.5 | <0.001 |
| Patients | 885 | 15 | 44.4 | 8.2 | 36 | 60 | | |
| Informal caregivers | 755 | 12 | 45.0 | 6.9 | 48 | 60 | | |
| **Aggression as dysfunctional or undesirable phenomenon** | | | | | | | 23.7 | <0.001 |
| Nurses | 782 | 6 | 25.3 | 4.1 | 30 | 30 | | |
| Patients | 885 | 6 | 23.7 | 5.3 | 30 | 30 | | |
| Informal caregivers | 755 | 6 | 24.4 | 4.2 | 30 | 30 | | |
| **Aggression as functional or comprehensible phenomenon** | | | | | | | 22.4 | <0.001 |
| Nurses | 782 | 4 | 15,0 | 3.6 | 20 | 20 | | |
| Patients | 885 | 4 | 14,0 | 4.1 | 20 | 20 | | |
| Informal caregivers | 755 | 4 | 13,9 | 3.5 | 20 | 20 | | |
| **Aggression as a protective measure** | | | | | | | 0.8 | 0.44 |
| Nurses | 782 | 2 | 6.8 | 2.2 | 10 | 10 | | |
| Patients | 885 | 2 | 6.7 | 2.5 | 10 | 10 | | |
| Informal caregivers | 755 | 2 | 6.7 | 2.2 | 10 | 10 | | |

physically. Nurses also agreed more often than the other two groups that aggression constitutes violence against nurses, is always negative and unacceptable and that feelings should be expressed in another way as it is a disturbing intrusion to dominate others. On the other hand, nurses were the group that most often disagreed that aggression can be the start of a positive nurse-patient relationship, that it is healthy reaction to feelings of anger, that it allows a better understanding of the patient's situation, and that aggression is a form of communication (See S1 Table).

A comparison of the total scores and sub-scores showed statistically significant differences between the three groups. First, based on the total score, nurses had more negative views and a lower tolerance toward patient aggression. Second, regarding the sub-scores, nurses perceived aggression as a 'dysfunctional or undesirable phenomenon' and a 'functional or comprehensible phenomenon' more often than patients and informal carers. No group differences were found in attitudes toward aggression as 'a protective measure' (Table 3).

S2 Table shows the results of the regression analysis of sociodemographic characteristics on attitudes toward aggression. Nurses had significantly higher total scores compared to the other groups, and significantly higher dysfunction and function sub-scores compared to patients, while informal caregivers had significantly higher scores related to dysfunction attitudes than patients. These results indicate that nurses had the lowest tolerance toward patient aggression, while patients had the highest tolerance toward aggression. In general, those over 51 years old had higher scores regarding the protection sub-score (aggression as a protection measure) comparing to those under 30 years old, indicating that older persons had more negative views and lower tolerance regarding aggression as a protective measure. Furthermore, females and those with a higher education level had higher total scores, and sub-scores regarding dysfunction, function and protection, indicating that these two groups had a lower tolerance toward aggression than males and those with secondary education and below, respectively (S2 Table).

S3 Table shows the results of the regression analysis of the sociodemographic characteristics on attitudes toward aggression separated by the participant group. Nurses in the age group 31–40 years had a higher score than other age groups regarding aggression as a protective measure. Female nurses had significantly higher total scores as well as dysfunction and function sub-scores than males. Nurses with 16–20 years of working experience had significantly higher

total scores and scores related to dysfunction attitudes. In addition, those nurses with the shortest amounts of working experience had significantly higher scores pertaining to aggression as a positive expression. Further, nurses who had participated in aggression management training Levels I and II had significantly higher scores in positive expression and protective measure, while nurses who had participated in all three levels of aggression management training had higher total scores and dysfunctional/undesirable sub-scores.

Female patients had significantly higher total scores and regarding attitudes toward aggression as dysfunction/undesirable sub-scores. Informal caregivers who were over 51 years old had significantly higher scores in attitudes of aggression as a protective measure. In addition, informal caregivers who were receiving disabled living allowance had significantly higher scores for the dysfunctional/undesirable sub-scale (S3 Table).

Based on the sensitivity analysis, no differences in the results were found (S4 Table).

## Discussion

In this cross-sectional study, we examined and compared attitudes toward patient aggression in psychiatric hospitals, and identified factors associated to these attitudes. The current study is unique in its comparison of attitudes toward aggression between three different groups. Nurses' high scores in the dysfunctional domain of the POAS-S is congruent with previous studies that have reported that nurses view aggression as a harmful, offensive and destructive behavior in patients [15, 57, 58]. This finding may reflect the traditional medical/psychiatric perspectives in which aggression is seen as undesirable, an illness-related behavior [59], based on patients' own responsibility, and resulting from negative emotions toward staff and co-patients [58]. The results may also reveal that when on the frontline in psychiatric hospitals, nurses are more likely to be confronted with aggressive events due to prolonged direct contact with patients in the course of care [60, 61]. In other words, if nurses encounter aggressive behavior more frequently than informal care givers or patients, it may explain their less tolerant views toward aggressive situations in health care settings. Understanding nurses' attitudes is therefore key as nurses' ambivalence regarding the acceptance of aggression and appropriate attitudes toward patient aggression may reflect how they manage patient aggression (i.e. coercive versus de-escalating measures) [35].

On the other hand, health care staff in this survey was the only group of respondents who had been trained to manage patients' aggressive behavior in inpatient psychiatric services. At the time of the data collection a great majority of staff members (88–74%) had participated in aggression management training, which targeted understanding aggression from different perspectives, and offered hands-on skills in how to protect against patient attacks. Nurses who had participated in all three levels of the training (Levels I–III) had still significantly less tolerant attitudes toward patients despite the general clinical assumption that education and training programs for staff reduce or eliminate patient aggressive behavior at work. Some previous studies have still shown that training could produce at least short-term positive improvements in nurses' attitudes toward patient aggression [62, 63]. However, a recent Cochrane review showed that the evidence is very uncertain about the short-term effects of education and training on aggression, while on the long run, education may not reduce aggression at all, compared to no intervention [64]. It has also been found that staff training related to the management of patient aggression can be even harmful and cause injuries [65]. Therefore, the use of training for changing attitudes should be carefully considered as it remains unclear how attitudes could be changed to be more tolerant [29, 47, 66].

Females and those with a higher education level in our study had higher total scores and sub-scores regarding dysfunction, function and protection, indicating that females and those

with a higher education level had lower tolerance toward aggression than males and those with secondary education and below. This result is consistent with a study reported by Jansen et al. [15], which found that female nurses perceived aggression as a destructive phenomenon more commonly than their male colleagues did. In a study conducted in Japan, the authors speculated that female nurses who faced patient aggressive behavior may concentrate on their own distress, which hinders their ability to respond to external attacks [67]. If patient aggression events are typically handled by male nurses with a lower education level, and more educated nurses are responsible in administrative tasks at the nurses' station, female nurses' may lack the experience and skills needed to handle these challenging situations. More research is therefore needed to understand nurse attitudes and management styles of patient aggressive events at the ward level in Asia.

We found both positive and negative views toward aggression, which may influence the adoption of person-centered approaches or the use of containment measures, respectively [17]. Opposite to Whittington's study [24], we found that nurses over 51 years old had more negative views and lower tolerance regarding aggression as a protective measure compared to those under 30 years old. Previous studies have also found negative association between the length of work experience and a positive attitude toward aggression [19, 23]. Verhaeghe et al. [13] proposed, based on their results, that more experienced nurses may lose a positive perspective and tolerance toward aggression. This development over time toward a tendency to place blame can be explained by the possible impact of patient aggression on nurses. The authors also found that burnout and post-traumatic stress increased significantly for mental health nurses employed more than 10 years. If these assumptions are valid, our finding is crucial from the point of view of clinical practice. As role models are central to attitude formation [15], it is important to ensure that older nurses with longer work experience receive support in managing their own emotions and attitudes toward patients in psychiatric wards to avoid negative role modelling for young and less experienced nurses.

Typically, studies on attitudes toward aggression have focused on patients and staff only. Although relevant, this approach has ignored informal caregivers' perceptions. Overlooking an inter-subjective approach between nurses and patients may have missed important information about how informal caregivers perceive patient aggression. Our results suggest that young informal caregivers may be in the most vulnerable position with regard to experiencing aggression. Previous studies have identified that informal caregivers do not necessary speak about the patient-initiated violence they may have faced [68]. It has also been assumed that family members may have hostility and other negative feelings toward patients due to aggressive experiences. Therefore, it is important to ask about safety at home when informal caregivers are visiting a patient on the ward [69]. Interventions to support informal caregivers' positive coping strategies toward aggressive behavior [28] and supporting them in identifying early warning signs of patient aggression [69] could be helpful to alleviate family burden and distress.

## Strengths and limitations

Regarding the strengths of this study, we compared attitudes between three groups, including informal caregivers', whose perceptions are often neglected in comparative studies. The sample size was sufficiently powered to conduct the group comparisons. Even so, limitations of the current study need to be taken into account when interpreting the results. First, this study used a convenience sample as only those participants who showed interested in joining the study were recruited. Therefore, we are unsure how biased the data are toward more positive attitudes. Second, we collected the data using a self-report tool, which may have affected the

likelihood for participants to respond in a socially desirable manner. Third, this study did not include other potentially important variables to describe factors related to attitudes or describe clinical practice. In the future, organizational-level outcomes, such as the use of coercive methods, patient and family members' complaints and nurse-patient ratios could shed light on this issue.

## Implications

Our study findings have implications for clinical practice, training, research, and mental health policy. First, our study showed that as attitudes toward aggressive behavior vary between different stakeholder groups, it is necessary to critically evaluate if current aggression management practices are meeting the needs and expectations of different stakeholders. A limited number of comparative surveys between nurses, patients and informal caregivers may subsequently lead to biased practices of nurses' perceptions only. Our findings suggest that using multi-approach surveys are an effective way to reveal potential mismatches in care delivery and expectations of care recipients. We also found that respondents' attitudes toward aggression varied in the nurses' data regarding their age: those over 51 years old had more negative attitudes and less tolerant attitudes in terms of aggression being used as a protection measure, compared to those under 30 years old. As social learning is a powerful mechanism of the socialization process in psychiatric settings in understanding which behaviors are appropriate and which ones are not, [15] younger nurses in our data may not have yet adopted negative attitudes toward patient aggression. Continual support is needed to avoid the transmission of a negative caring culture to new nursing generations, to maintain ethically high standards of nursing practice and skills, and to solve both ethical and moral conflicts arising from daily clinical work [70]. These findings merit attention in other clinical areas in which health and social care staff face patient aggression in their daily work.

The need for support is not only regarding aggression management but also for the emotional regulation of nurses [71]. To increase nurses' understanding of the nature of aggressive behavior, which thus leads to more empathetic attitudes [13], nurses should be more aware of the possible proactive and functional nature of aggression. If understanding aggression leads to better work alliances, it might improve the quality of clinical practice as well. It is therefore important that working environments are made more suitable and that support systems are in place for nurses to help them to manage their feelings and perceptions in an emotionally demanding work.

Further studies should be conducted to investigate levels of anger and other emotions in nurses who have witnessed patient aggressive behavior, and to find out whether these emotion levels are linked with nurses' attitudes and coercive practices used on hospital wards. This could offer a clearer picture of the impact of nurses' emotions and attitudes on nursing practice. Future research should also include more objective data collection methods that combine subjective and objective factors in predicting attitudes and their impact on daily practices. Also, more studies should focus on how to improve the mechanism of patients' own aggression management. In addition, to ensure the generalizability of the results, future research could be replicated in different psychiatric hospitals and this research could be extended to include patient groups in other clinical fields as well as other stakeholders.

## Supporting information

**S1 Table. Comparison of nurses', patients', and informal caregivers' perceptions of aggression.**
(DOCX)

**S2 Table. Results of regression analysis for the total sample.**
(DOCX)

**S3 Table. The results of the regression analysis separately for each participant group.**
(DOCX)

**S4 Table. Sensitivity analysis to identify possible multicollinearity in the regression model.**
(DOCX)

## Acknowledgments

The research team would like to express deep gratitude to the following people who contributed to this project: Ms LF Wong and Mr MH Chow for their support and assistance in this project; and our hospital partners and staff, for their help in facilitating the data collection. Special thanks to all the participants who generously shared their valuable time and experience for the purposes of this project.

## Author Contributions

**Conceptualization:** Maritta Välimäki, Tella Lantta.

**Data curation:** Joyce Lam.

**Formal analysis:** Lee Paul.

**Funding acquisition:** Maritta Välimäki.

**Investigation:** Maritta Välimäki, Joyce Lam.

**Methodology:** Maritta Välimäki.

**Project administration:** Maritta Välimäki.

**Resources:** Maritta Välimäki.

**Software:** Joyce Lam, Lee Paul.

**Supervision:** Maritta Välimäki.

**Validation:** Joyce Lam.

**Visualization:** Maritta Välimäki.

**Writing – original draft:** Maritta Välimäki, Tella Lantta.

**Writing – review & editing:** Joyce Lam, Daniel Bressington, Teris Cheung, Wai Kit Wong, Po Yee Ivy Cheng, Chi Fai Ng, Tony Ng, Chun Pong Yam, Glendy Ip, Lee Paul.

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
