## [Decision Letter · Decision Letter 0]

17 May 2022

PONE-D-21-29009Nurses’, patients’, and informal caregivers’ attitudes toward aggression in psychiatric hospitals: a comparative survey studyPLOS ONE

Dear Dr. Välimäki,

Thank you for submitting your manuscript to PLOS ONE. After careful consideration, we feel that it has merit but does not fully meet PLOS ONE’s publication criteria as it currently stands. Therefore, we invite you to submit a revised version of the manuscript that addresses the points raised during the review process.

Your manuscript has been assessed by an expert reviewer, whose comments are appended below. The reviewer has made some important points about several aspects of the methodology, as well as the framing of the results and conclusions, which you should address carefully in your revised manuscript. Please note that we have only been able to secure a single reviewer to assess your manuscript. We are issuing a decision on your manuscript at this point to prevent further delays in the evaluation of your manuscript. Please be aware that the editor who handles your revised manuscript might find it necessary to invite additional reviewers to assess this work once the revised manuscript is submitted. However, we will aim to proceed on the basis of this single review if possible. 

We look forward to receiving your revised manuscript.

Kind regards,

Joseph Donlan

Editorial Office

PLOS ONE

**Journal requirements:**

“The research team would like to express deep gratitude to the following people who contributed to this project: Ms LF Wong and Mr MH Chow for their support and assistance in this project; and our hospital partners and staff, for their help in facilitating the data collection. Special thanks to all the participants who generously shared their valuable time and experience for the purposes of this project. MV has received funding for this study: the Start-up funding by the Hong Kong Polytechnic University (grant name: Developing user-centered treatment culture to prevent patient aggressive events in psychiatric hospitals, https://www.polyu.edu.hk/en/) and the Academy of Finland fund (grant numbers: 294298, 307367, https://www.aka.fi/en/). The funders had no role in study design, data collection and analysis, decision to publish, or preparation of the manuscript.**”**

“MV has received funding for this study: the Start-up funding by the Hong Kong Polytechnic University (grant name: Developing user-centered treatment culture to prevent patient aggressive events in psychiatric hospitals, https://www.polyu.edu.hk/en/) and the Academy of Finland fund (grant numbers: 294298, 307367, https://www.aka.fi/en/). The funders had no role in study design, data collection and analysis, decision to publish, or preparation of the manuscript.”

6. Please amend your list of authors on the manuscript to ensure that each author is linked to an affiliation. Authors’ affiliations should reflect the institution where the work was done (if authors moved subsequently, you can also list the new affiliation stating “current affiliation:….” as necessary).

Reviewers' comments:

Reviewer's Responses to Questions

**Comments to the Author**

1. Is the manuscript technically sound, and do the data support the conclusions?

Reviewer #1: Partly

2. Has the statistical analysis been performed appropriately and rigorously? 

Reviewer #1: No

3. Have the authors made all data underlying the findings in their manuscript fully available?

Reviewer #1: Yes

4. Is the manuscript presented in an intelligible fashion and written in standard English?

Reviewer #1: Yes

5. Review Comments to the Author

Reviewer #1: Thank you for the opportunity to review. Aggression is a major and important issue in mental health care. I agreed with the authors’ perspective to understand patients’ aggression from multiple angles but I thought it is necessary to modify or add some descriptions.

Abstract

Do authors present the results of the regression analysis or logistic analysis? I think it is better to clarify what analysis was based on the results and describe the ORs or �eta including 95%CIs.

BACKGROUND

Authors said, ‘attitudes towards aggression plays a crucial role in how aggressive incidents are seen, occur or are managed in health care settings.’ However, the relationships between attitude toward aggression and other factors are not clear. I think further explanation about the relationships is needed. In addition, what is the definition of attitude? In the scale name of POAS, it is used the word ‘perception’. It seems that they are different concepts. Would you tell me the reasons that it is appropriate to use ‘attitude’ as perception?

Although authors predicted that nurses would have most negative and less tolerant attitudes toward aggression, what is the basis for the hypothesis? I think it is necessary to add the explanation of why the hypothesis was led with references.

MATERIALS AND METHODS

In Table 1, the number of admissions is shown in a year, but the number of discharged patients is shown in a month. I think it is better to present the periods consistently.

Although the nurse participants group included assistant nurses, is it appropriate to regard assistant nurses as the same members of the professional group? It is considered that the registered and enrolled nurses have the expertise and the experience of the care using them. Do Assistant nurses also have them? If so, the authors have to add an explanation of the nursing qualification system in Hong Kong. Or if not, you have to add the reason including assistant nurses. In addition, is the training for violence management for nurses only, not including assistant nurses?

The definition of informal caregivers is family members, relatives, and friends. However, I wonder how the relatives and friends did regard as caregivers. If the authors had the other inclusion criteria in this study, it is necessary to add this.

The authors need to describe how missing values were handled.

RESULT

The length of work experience was categorized into 5 groups in the method section. However, in the result section, it seems to be categorized into 4 groups. Which is the categorization correct?

In the nurses' group, there may be a correlation strongly between age and work experience. I think the authors have to explain how the authors considered and analyze the multicollinearity.

I think it is better to add the results about the fitness of the models such as the adjusted R-squared in Table 5 and the result of the Hosmer-Lemeshow test or the Nagelkerke’s R squared in Table 6.

Discussion

What does the sentence mean, ‘The finding is interesting when compared with those reported previously showing that training could produce at least short-term positive improvements in nurse attitudes towards patient aggression’? Does it mean that the findings present the long-term impacts of training, or these findings did not present the impacts of training on attitude toward aggression? It is necessary to describe the interpretation of these findings clearer.

It is understandable that female nurses had lower tolerance toward aggression than males. I think it is because of not only the lack of experience and skills but also the differences in biological characteristics. Therefore, I think it may be difficult to suggest the lack of skill simply and it may be also important to build a suitable working environment and support system.

The nurses’ length of working experience was categorized into five groups. But this categorization seems to make the interpretation of the results difficult. Although the authors described the possible impacts of age, there are no explanations about working experience. How did the author consider the result that only one group with experience of 16-20 years had affected aggression?

6. PLOS authors have the option to publish the peer review history of their article (what does this mean?). If published, this will include your full peer review and any attached files.

Reviewer #1: **Yes: **Ryo Odachi

---

## [Author Response · Author response to Decision Letter 0]

17 Aug 2022

Please see the attachment 'Response to reviewers'.

---

## [Editor Report · Decision Letter 1]

31 Aug 2022

Nurses’, patients’, and informal caregivers’ attitudes toward aggression in psychiatric hospitals: a comparative survey study

PONE-D-21-29009R1

Dear Dr. Välimäki,

We’re pleased to inform you that your manuscript has been judged scientifically suitable for publication and will be formally accepted for publication once it meets all outstanding technical requirements.

Kind regards,

Sónia Brito-Costa, Ph.D.

Academic Editor

PLOS ONE
---

## [Editor Report · Acceptance letter]

6 Sep 2022

PONE-D-21-29009R1 

Nurses’, patients’, and informal caregivers’ attitudes toward aggression in psychiatric hospitals: a comparative survey study 

Dear Dr. Välimäki:

I'm pleased to inform you that your manuscript has been deemed suitable for publication in PLOS ONE. Congratulations! Your manuscript is now with our production department. 

Kind regards, 

on behalf of

Dr. Sónia Brito-Costa 

Academic Editor

PLOS ONE